

# Honey bees and bumble bees occupying the same landscape have distinct gut microbiomes and amplicon sequence variant-level responses to infections

Navolle Amiri[1,*], Mia M. Keady[1,2,3,*] and Haw Chuan Lim[1,3]

[1] Department of Biology, George Mason University, Fairfax, VA, United States
[2] Nelson Institute for Environmental Studies, University of Wisconsin—Madison, Madison, WI, United States
[3] Center for Conservation Genomics, Smithsonian's National Zoo and Conservation Biology Institute, Washington, D.C., United States
* These authors contributed equally to this work.

Corresponding author
Haw Chuan Lim, hlim22@gmu.edu

## ABSTRACT

The gut microbiome of bees is vital for the health of their hosts. Given the ecosystem functions performed by bees, and the declines faced by many species, it is important to improve our understanding of the amount of natural variation in the gut microbiome, the level of sharing of bacteria among co-occurring species (including between native and non-native species), and how gut communities respond to infections. We conducted 16S rRNA metabarcoding to discern the level of microbiome similarity between honey bees (*Apis mellifera*, N = 49) and bumble bees (*Bombus* spp., N = 66) in a suburban-rural landscape. We identified a total of 233 amplicon sequence variants (ASVs) and found simple gut microbiomes dominated by bacterial taxa belonging to *Gilliamella*, *Snodgrassella*, and *Lactobacillus*. The average number of ASVs per species ranged from 4.00–15.00 (8.79 ± 3.84, mean ± SD). Amplicon sequence variant of one bacterial species, *G. apicola* (ASV 1), was widely shared across honey bees and bumble bees. However, we detected another ASV of *G. apicola* that was either exclusive to honey bees, or represented an intra-genomic 16S rRNA haplotype variant in honey bees. Other than ASV 1, honey bees and bumble bees rarely share gut bacteria, even ones likely derived from outside environments (*e.g.*, *Rhizobium* spp., *Fructobacillus* spp.). Honey bee bacterial microbiomes exhibited higher alpha diversity but lower beta and gamma diversities than those of bumble bees, likely a result of the former possessing larger, perennial hives. Finally, we identified pathogenic or symbiotic bacteria (*G. apicola*, *Acinetobacter* sp. and *Pluralibacter* sp.) that associate with Trypanosome and/or *Vairimorpha* infections in bees. Such insights help to determine bees' susceptibility to infections should gut microbiomes become disrupted by chemical pollutants and contribute to our understanding of what constitutes a state of dysbiosis.

## INTRODUCTION

There is a growing appreciation of the importance of the symbiotic microbiome, specifically the gut microbiome. The microbiome consists of all microorganisms which colonize regions within or upon a host organism. In animals, symbiotic host-microbiome relationships have been shown to impact host evolution and physiology (*Fraune & Bosch, 2010*). While studies on gut microbiota, especially gut bacterial and archaea microbiomes, have accelerated in recent years due to technological advances (*Bahrndorff et al., 2016*; *West et al., 2019*) much about the diversity and variation of gut microbiota in wild and naturalized species remains to be explored. Many studies focus on insects because of their ecosystem and economic importance (*Engel & Moran, 2013*; *Arora et al., 2022*). Moreover, they are one of the most diverse groups of animals, occupying many environments and filling many ecological niches (*Basset et al., 2012*). This ecological diversity is due in part to insects maintaining symbiotic relationships with their gut microbiota, which aid their hosts in a wide range of metabolic and physiological processes such as digestion and defense against and pathogens and parasites (*Engel & Moran, 2013*).

Among insect functional groups, one of the most important are pollinators. Pollinator species are essential in maintaining human food production as well as aiding in plant reproduction. Economically, insect pollinators are especially important as they are needed for around 75% of crops consumed by humans (*Klein et al., 2007*). Thus, there is a growing need to understand the contribution of microbiota, especially gut bacterial microbiota, to the health and ecological well-being of pollinator populations. In the United States, the primary type of insect pollinators used in agriculture is the western honey bee (*Apis mellifera*), with other species of bees used in more exclusive contexts (*Potts et al., 2010*). For example, there is a greater dependency on bumble bees (*Bombus* spp.) for crops such as cranberries and blueberries (*Vaccinium* spp.) as they are able to forage in colder, windier weather (*Calderone, 2012*; *Martin, Fountain & Brown, 2019*). Both honey and bumble bees are classified as eusocial bees, although there are distinct differences in life history. Honey bee colonies typically contain around 35,000 bees, with queens being able to live up to several years (*Winston, 1991*; *McMenamin et al., 2018*). Bumble bees live in smaller annual colonies and go through both social and solitary phases (*Goulson, 2003*; *McMenamin et al., 2018*). These differences among species are likely to affect the transmission of gut bacterial microbiota across generations and among populations (*Kwong et al., 2017*).

Previous studies have shown that honey bee bacterial microbiomes are highly conserved, with nine species making up 95% of the gut bacterial taxa, as determined by 16S rRNA sequencing (*Kwong & Moran, 2016*; *Horak, Leonard & Moran, 2020*). Despite having diverged from honey bees around 80 million years ago, bumble bee species share most of their core gut bacterial genera with honey bees, indicating the functional importance of these bacterial taxa in the gut environments of eusocial bees (*Lim et al., 2015*; *Kwong et al., 2017*). The development of core microbiomes of honey and bumble bees through their life cycles are strongly determined by their social behaviors (*Kwong et al., 2017*). For example, newly emerged honey bees developed typical gut microbiota through trophallactic interactions with other bees (*e.g.*, nurse bees) as well as the hive

environment itself (*Martinson, Moy & Moran, 2012*; *Kwong et al., 2017*; *Bulson et al., 2021*). In bumble bees, gut microbiome of the first batch of workers after colony establishment closely resemble those of the founding queen, indicating her importance in the vertical transmission of bacteria (*Hammer et al., 2021*; *Su et al., 2021*). As the bumble bee colony ages, within- and between-colony heterogeneity in microbiome decreases, suggesting that selective forces act to maintain a recurring and typical set of taxa within a bee species (*Su et al., 2021*).

Recently, scientists have also begun to look at how the bee gut bacterial microbiome may provide protective functions against enteric parasites (*Goulson et al., 2015*; *Mockler et al., 2018*). Parasitic infections in bees have increased in recent years due to the increased deployment of domesticated honey and bumble bees in farms or greenhouses, and the subsequent spill-over of pathogens from managed bees into wild bees (*Martin, Fountain & Brown, 2019*). Two of the most common groups of parasites affecting bees are *Vairimorpha* spp. (*e.g.*, *Vairimorpha bombi* and *V. cerenae*) and the protozoan Trypanosomes (*e.g.*, *Crithidia bombi*, *C. expoeki*, *C. mellificae*, *Lotmaria passim*, and *Leptomonas scantii*) (*Cariveau et al., 2014*; *Hristov et al., 2020*). In honey bees, *Vairimorpha* infections can negatively impact brood rearing, pollen collecting activities, and overall longevity of adult bees (*Botías et al., 2013*; *Hristov et al., 2020*). The trypanosome *Crithidia bombi* can increase mortality, decrease the success of colony-founding by queens, and reduce colony size and fitness in multiple species of bumble bees (*Mockler et al., 2018*).

The goals of our current study are to understand how gut bacterial microbiomes of honey bees and bumble bees vary across individuals and sites within a rural-urban landscape, and how the bacterial gut microbiome may be related to infection by parasites. In spite of the importance of honey bees and bumble bees in agriculture and uncultivated ecosystems, few studies have been conducted to quantify diversity of and similarity in gut microbial communities and sharing of bacterial ASVs between the two groups of bees when they occupy the same environments. Here, we characterized gut microbiomes of honey bees (*A. mellifera*) and eight bumble bee (*Bombus*) species collected from six sampling sites, and further conducted statistical diversity analyses on three locally abundant focal species (*A. mellifera*, *B. griseocollis*, and *B. impatiens*). Because of differences in colony size, social structure, and general life history between honey bees and bumble bees, we hypothesize that they differ with respect to gut microbiome diversity at the alpha (the number of bacterial taxa per bee), beta (the turnover in bacterial species from one individual bee to the next) and gamma (overall accumulated diversity within the study region) levels (*Whittaker, 1972*). Additionally, given recent studies showing the protective functions of gut microbiome against infection by enteric parasites (*Vairimorpha* spp. and Trypanosomes) (*Koch & Schmid-Hempel, 2011*; *Mockler et al., 2018*; *Horak, Leonard & Moran, 2020*), and with infections being correlated with altered gut microbiome (*Koch & Schmid-Hempel, 2011*; *Cariveau et al., 2014*; *Goulson et al., 2015*), we hypothesize that specific bacterial taxa are associated with infected and non-infected bees.

**Table 1 Number of bees collected from the various study sites in northern Virginia.**

| | Blandy Experimental Farm | The Clifton Institute | Piedmont Environmental Council | St. Benedict Monastery | George Mason University – Fairfax Campus | Residence— Winchester VA | Total sample size by species |
|---|---|---|---|---|---|---|---|
| Latitude, Longitude | 39.06, −78.06 | 38.77, −77.80 | 38.71, −77.79 | 38.75, −77.56 | 38.83, −77.31 | 39.22, −78.23 | |
| *A. mellifera* | 2 | 12 | 9 | 17 | 0 | 9 | 49 |
| *B. auricomus* | 0 | 1 | 0 | 1 | 0 | 0 | 2 |
| *B. bimaculatus* | 0 | 1 | 0 | 1 | 0 | 0 | 2 |
| *B. fervidus* | 0 | 0 | 0 | 0 | 3 | 0 | 3 |
| *B. griseocollis* | 4 | 8 | 0 | 3 | 3 | 3 | 21 |
| *B. impatiens* | 9 | 10 | 0 | 11 | 5 | 0 | 35 |
| *B. pensylvanicus* | 0 | 0 | 0 | 0 | 1 | 0 | 1 |
| *B. perplexus* | 0 | 2 | 0 | 0 | 0 | 0 | 2 |
| Total sample size by site | 15 | 34 | 9 | 33 | 12 | 12 | 115 |

## MATERIALS AND METHODS

### Bee collection

Bee collection occurred during late July to early August in 2018 and mid-June of 2019. Bumble bees were caught in the field with a hand net while honey bees were either caught in the field or from hives. A total of 115 individuals from eight bee species (*Apis mellifera*, *Bombus auricomus*, *B. bimaculatus*, *B. fervidus*, *B. griseocollis*, *B. impatiens*, *B. pensylvanicus*, and *B. perplexus*) were collected from six sites in northern Virginia (Table 1). Permissions to access these sites were obtained from B. Harris (The Clifton Institute), T. Roulston (Blandy Experimental Farm), C. Vuocolo (Piedmont Environmental Council), L. Edsall (St. Benedict Monastery), G. Perilla (George Mason University) and S. Brandsetter (private residence, Winchester, VA). Bumble and honey bee species were identified by T. Roulston (Dept. of Environmental Sciences, University of Virginia). Bumble bees that occurred locally can be differentiated by color pattern except for *B. auricomus* and *B. pensylvanicus* (*Williams et al., 2014*). Individuals of these two species were differentiated through a morphological feature, tibia spine (*Williams et al., 2014*). Collected bees were first kept individually in 50 ml falcon tubes, and then moved into 2 ml centrifuge tubes within 15 min. These tubes were then placed into a tank of liquid nitrogen (LN2), transported to George Mason University, and stored in a −20 °C freezer until the initiation of laboratory work. Although insects are not under the purview of our institutional animal care and use guidelines, the rapid freezing of bees in LN2 ensures rapid euthanization.

### Molecular methods and sequencing

#### DNA extraction and 16S rRNA amplification

We sterilized outer surfaces of bees using a 6% bleach-water mixture and then washed with distilled water to remove microbes found on the surface of the body. The heads were

removed and stored in −80 °C for archiving while abdomens were placed in 1.5 ml tubes with two ¼ inch ceramic beads (MP Biomedicals, Santa Ana, CA, USA) for DNA extraction. Using the Fastprep-24 Classic Sample Preparation System (MP Biomedicals, Santa Ana, CA, USA), we homogenized each bee abdomen by shaking the tubes that contained lying matrix for 20 s twice. To perform DNA extraction, we used the PureLink™ Genomic DNA Mini Kit (Thermo Fisher Scientific, Waltham, MA, USA) in accordance with the manufacturer's protocol (batch size: 16–24 samples extracted at a time). For each individual, the V3–V5 region of the 16S rRNA bacterial gene was amplified using a forward primer 515F-Y-I2S (TCGTCGGCAGCGTCAGATGTGTATAAGAGACAGGTG YCAGCMGCCGCGGTAA) and reverse primer 939R-I2S (GTCTCGTGGGCTCGGAG ATGTGTATAAGAGACAGCTTGTGCGGGCCCCCGTCAATTC) (adapted from *Muletz-Wolz, Fleischer & Lips, 2019*). These were fusion primers with tails that enabled incorporation of Illumina sequencing adapters during a subsequent round of indexing PCR. Given that bees were subjected to surface sterilization, we anticipate that most of the bacteria we sequence would have derived from the gastrointestinal tract.

We set up polymerase chain reactions (PCRs) on ice and preparation of reactions occurred within a sterile PCR laminar flow hood. We prepared 10 μL reactions, each containing 5.32 μL water, 2 μL 5X Q5 Reaction Buffer, 0.2 μL dNTP (10 mM each), 0.3 μL 10 mM forward primer, 0.3 μL 10 mM reverse primer, 0.6 μL BSA, 0.08 μL Q5 Taq polymerase (New England Biolabs, Ipswich, MA, USA) and 1.2 μL of template DNA. A negative control was included in each batch of PCR. Thermal cycling started at 98 °C for 30 s for 1 cycle, then 25 cycles of 98 °C for 10 s, 62 °C for 15 s, 72 °C for 15 s, and finally one cycle at 72 °C for 2 min. Upon completion of PCRs, we ran the products on a 1.5% agarose gel and verified if PCR products were present. For each bee, we conducted two separate 10 μl PCRs. Duplicate samples were combined and cleaned with 1.5x volume Seramag Speedbeads and two 200 μl 70% ethanol washes (Thermo Fisher Scientific, Waltham, MA, USA) (*Rohland & Reich, 2012*). Next, we conducted second-round PCRs where we incorporated unique 8 bp Illumina i5 and i7 indexes and sequencing adapters to each sample. We performed these indexing PCRs in 25 μL reactions made with 9.25 μL of water, 0.5 μL of dNTP, 5 μL of 5X Q5 Reaction Buffer, 0.25 μL Q5 Taq polymerase, 3 μL of i5 primer, 3 μL of i7 primer, and 4 μL of first-round PCR products. PCR conditions were 98 °C for 30 s, followed by 8 cycles of 98 °C for 10 s, 62 °C for 15 s, 72 °C for 30 s, and 1 cycle at 72 °C for 2 min. These final products were again cleaned using Seramag Speedbeads.

### Sequencing & data processing

Concentration of cleaned final PCR products were quantified using a Qubit 3.0 Fluorometer (Invitrogen, ThermoFisher Scientific, Waltham, MA, USA). Next, we pooled all indexed PCR products in equimolar proportions. Finally, we sequenced the pool using the MiSeq Reagent Kit version 3 (2 × 300 bp) on an Illumina MiSeq sequencer at the Department of Biology, George Mason University, together with samples from other studies. Bee samples comprised ~5% of the MiSeq flowcell. This level of sequencing effort

was determined to be sufficient given that past studies have demonstrated that honey and bumble bees possess simple gut bacterial communities (*Martinson et al., 2011*).

Data processing and statistical analyses were conducted in RStudio (v. 1.2.1335) using R-3.6.1 (*R Development Core Team, 2019*) following analysis workflows based on *Muletz-Wolz, Fleischer & Lips (2019)* and *Keady et al. (2021)*. We used the DADA2 pipeline (v1.14.1) (*Callahan et al., 2016*) to quality filter data and identify amplicon sequence variants (ASVs), which were single exact sequences used to replace Operational Taxonomic Units (OTUs) (*Callahan, McMurdie & Holmes, 2017*). A phylogenetic tree was made with the FastTree package (*Price, Dehal & Arkin, 2009*) and taxonomic classifications were assigned using the Ribosomal Database Project naïve Bayesian classifier (*Wang et al., 2007*). Next, we merged the ASV table, taxonomy table, meta-data information, and phylogenetic tree into a phyloseq object (*McMurdie & Holmes, 2013*). The frequency method in the decontam R package (v. 1.1.2) was used to filter out potential contaminants (*Davis et al., 2018*). We also filtered out singleton ASVs, and ASVs classified as Cyanobacteria as the latter likely were sequenced derived from pollen. To correct for biases from unequal sequencing depths (20.54-fold difference in sequencing depth across sequenced bees), we rarified each sample to the same number of reads using the *rarefy_even_depth* function in the phyloseq R package (*McMurdie & Holmes, 2013*). We considered the limitations of rarifying samples when estimating alpha diversities (*McMurdie & Holmes, 2014*; *Willis, 2019*), but given that the ASV accumulation curves of most microbiome samples have plateaued (Fig. S1), we believe that there were few undetected taxa, and as such estimates of diversity metrics were not biased by rarefactions.

## Data analysis

### Microbiome diversity and differences

To identify common bacterial species, we determined ASVs that have high frequency of occurrence and relative abundance in bees of each species. We then assessed alpha and beta diversities to characterize microbiome richness of bee species and differences among individuals and species. Statistical analyses were restricted to the three focal bee species with the largest sample sizes (*Apis mellifera* $n = 49$, *Bombus griseocollis* $n = 21$, and *Bombus impatiens* $n = 35$). For alpha diversity, we calculated bacterial ASV richness and Faith's phylogenetic diversity of individual bees. Faith's phylogenetic diversity is determined by the sum of the total tree branch lengths connecting all bacterial ASVs present in a sample (*Faith, 1992*). For the focal species, we compared differences in alpha diversity using two ANOVAs (*aov* function; stats R package), following confirmation that normality assumption was met using Shapiro–Wilk normality (shapiro.test function; stats R package) and Levene's tests (leveneTest function; car package). Host species was the explanatory variable and the two alpha diversity measures were response variables. Tukey's honest significance test (*TukeysHSD* function; stats R package) was performed as a *post hoc* analysis to identify pairwise significant differences.

We used two beta diversity distance measures, Bray–Curtis and Jaccard, to characterize differences in microbial community composition among bees of individual species, and among bee species. To determine whether focal bee species have statistically distinct gut

bacterial communities, we conducted two PERMANOVAs with host species as an explanatory variable, site location as a covariate, and each of the two beta diversity measures as the response variable (*Oksanen et al., 2019*). Additionally, we tested significant differences in microbiome dispersion between host species with the *betadisper* function (*Oksanen et al., 2019*). Dispersion refers to the average variance in dissimilarity distances (Bray–Curtis and Jaccard) of individuals from the species/group centroid (*Anderson, Ellingsen & McArdle, 2006*; *Oksanen et al., 2019*).

Finally, we evaluated among-species differences in gamma diversity, which yield insights into the total diversity of each species by accounting for diversity within bees and turnover of microbial communities among individuals of the same species (*Whittaker, 1972*). To do this, we constructed ASV accumulation curves to determine the number of new ASVs that were discovered as more bees are sampled (*specaccum* function, method = "random"; vegan package).

### Indicator bacterial ASVs associated with parasite infection

For each focal bee species, we investigated the relationship between specific bacterial ASVs and infection by gut Trypanosome and/or *Vairimorpha* spp. parasites. We used DNA sequencing-based infection data for the same bees collected by another study (see *Lambrecht, 2021* for more details). We identified ASVs associated to infection status using the indicspecies package, function *mulitpatt* (max.order = 1) (*De Cáceres & Legendre, 2009*; *De Cáceres, Legendre & Moretti, 2010*). We adjusted $p$ values for multiple comparisons with function *p.adjust* and method "fdr". We report significant ASVs with a combined A and B IndVal score of at least 0.6 (*Dufrêne & Legendre, 1997*; *Van Rensburg et al., 1999*; *De Cáceres, Legendre & Moretti, 2010*). The IndVal score represents both specificity (*A score*: abundance of an ASV in an infection group) and fidelity (*B score*: frequency of the bacterial ASV across individuals in an infection group) (*Dufrêne & Legendre, 1997*) of associations. We assessed indicator bacterial ASVs in each of the focal bee species independently due to differences in their gut microbiome communities (Table 2). Due to low rates of *Vairimorpha* infection, *B. griseocollis* and *B. impatiens* were tested only for ASVs associated with Trypanosome infection or no infection. Trypanosomes parasites belonging to the genera of *Crithidia*, *Lotmaria*, and *Leptomonas* were identified by *Lambrecht (2021)*, but we did not differentiate them for our indicator species analyses due to limited infection by each Trypanosome type. In *A. mellifera*, we tested for ASVs associated with co-infection by *Vairimorpha* spp. and Trypanosomes, only by *Vairimorpha* spp., only by Trypanosomes, and no infection (Table 2).

## RESULTS

### Bacterial lineages

We surveyed the gut microbiome of 115 bees from eight different species, collected across six sites (Table 1). We obtained a total of 211,132 raw data sequence reads. The raw data was pre-processed to remove any contaminants and singletons. Any sequences shorter than 500 bp were also removed, leaving 191,982 reads post filtering. After rarifying, we

**Table 2 Parasitic infection rate of focal bee species by Trypanosome and/or *Vairimorpha*.** Low sample size and low infection rate of *Vairimorpha* in *Bombus* spp. restricted our indicator species analysis of *Bombus* spp. to two categories (no infection *vs.* infection by Trypanosome exclusively).

| Focal bee species | Co-infected | Trypanosome exclusively | *Vairimorpha* exclusively | No infection |
|---|---|---|---|---|
| *A. mellifera* | 7 | 10 | 5 | 27 |
| *B.griseocollis* | 2 | 14 | 1 | 4 |
| *B.impatiens* | 1 | 25 | 0 | 9 |

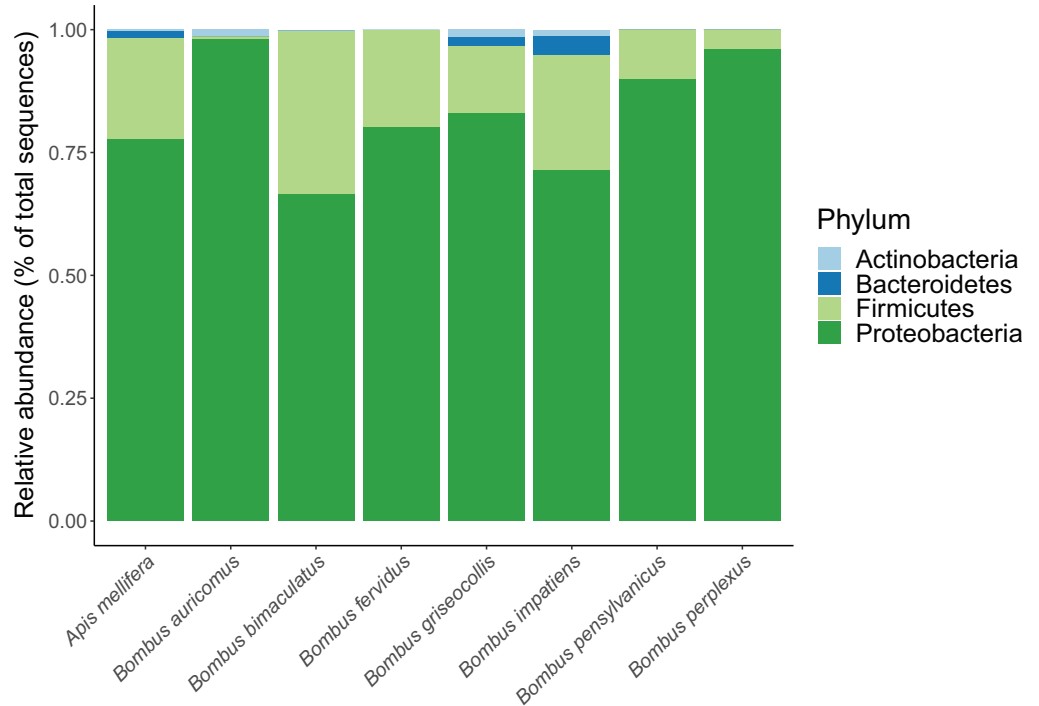

**Figure 1 Relative abundance of bacterial phyla in each bee species.** 

obtained a total of 60,835 high quality bacterial sequences with each sample having 529 reads (Fig. S1).

In total, we identified 233 unique ASVs from the following four bacterial phyla: Proteobacteria, Firmicutes, Bacteroidetes and Actinobacteria. Their overall relative proportions across focal species were 77.4%, 19.2%, 2.4% and 1.0%, respectively (Fig. 1; Table S1). *Apis mellifera* possessed a distinctive core microbiome with a pronounced pattern of ASV sharing among individuals. Five ASVs from three genera were shared by >80% of *A. mellifera* individuals, and these were *Lactobacillus apis* (ASV 6), *Lactobacillus* sp. (ASV 23), *Gilliamella apicola* (ASVs 1 and 7), and *Snodgrassella alvi* (ASV 3) (Fig. 2 Figs. S2 and S3). Their average relative abundances were 21.7% (ASV 3), 11.0% (ASV 1), 10.9% (ASV 6), 9.1% (ASV 7), and 3.1% (ASV 23). Unlike *A. mellifera*, we did not find dominant ASVs in *B. impatiens* or *B. griseocollis* individuals (*i.e.*, occurring in >80% of

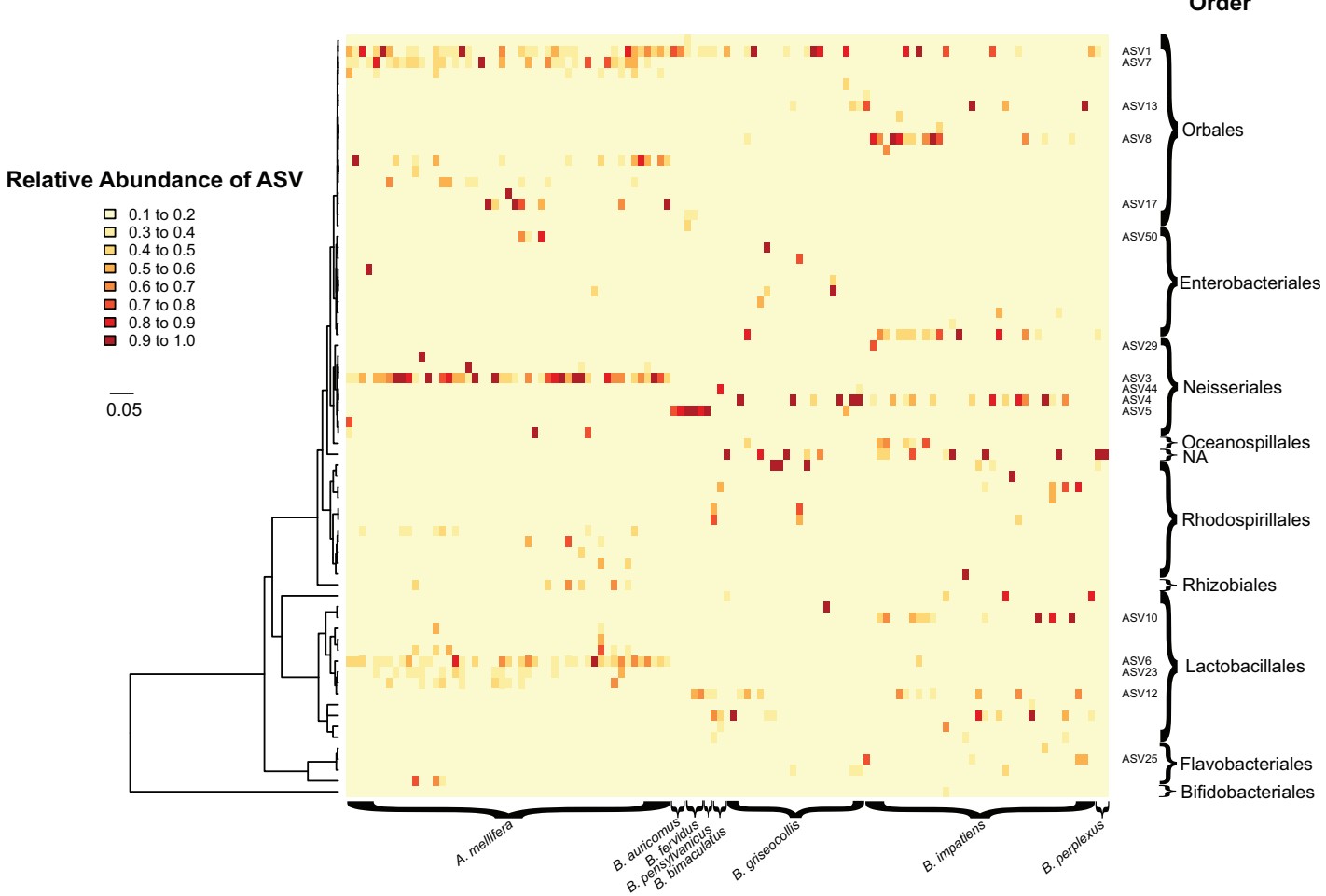

**Figure 2 Heatmap showing the relative abundance (0.1 to 1.0) of bacterial ASVs (rows) in individual bees (columns).** Columns are grouped based on the host bee species. Tree on the left of the heatmap indicates phylogenetic relationships between bacterial ASVs. Taxonomic orders of bacteria are shown on the right of the heatmap. Bacterial taxa common in focal species and mentioned in text are labeled by ASV numbers. Colors of heatmap indicate abundance classes of ASVs. *Apis mellifera* displays greater bacterial uniformity (*e.g.*, ASV 3) compared to bumble bees. ASV host specificity can also be observed, with some having higher abundances (*e.g.*, ASV 6) in one host species over another. Scale bar indicates the number of substitutions per nucleotide site along branches of the bacterial phylogenetic tree.

individuals of each species), the two focal bumble bee species with large sample sizes. Instead, the most common bacterial ASV in *B. impatiens* was present in only 68.6% of the bees (ASV 4, *Snodgrassella alvi*), and the remaining common bacteria (in 51.4–62.9% of samples) were: ASV 8 (*Orbus* sp.), *Gilliamella apicola* (ASV 1) and *Lactobacillus* sp. (ASV 10) (Fig. 2). The relative abundances of these ASVs were 10.5% (ASV 8), 9.8% (ASV 4), 7.9% (ASV 10) and 5.8% (ASV 1), when averaged across all *B. impatiens* individuals. In *B. griseocollis*, common bacteria, *Snodgrassella alvi* (ASV 4), *Gilliamella apicola* (ASV 1), *Bombiscardovia* sp. (ASV 52), and *Lactobacillus bombicola* (ASV 12), were present in 33.3–47.6% of samples. Their average relative abundances in *B. griseocollis* were 18.7% (ASV 4), 14.9% (ASV 1); ASV52 (1.1%) and ASV12 (2.6%) were prevalent but not abundant.

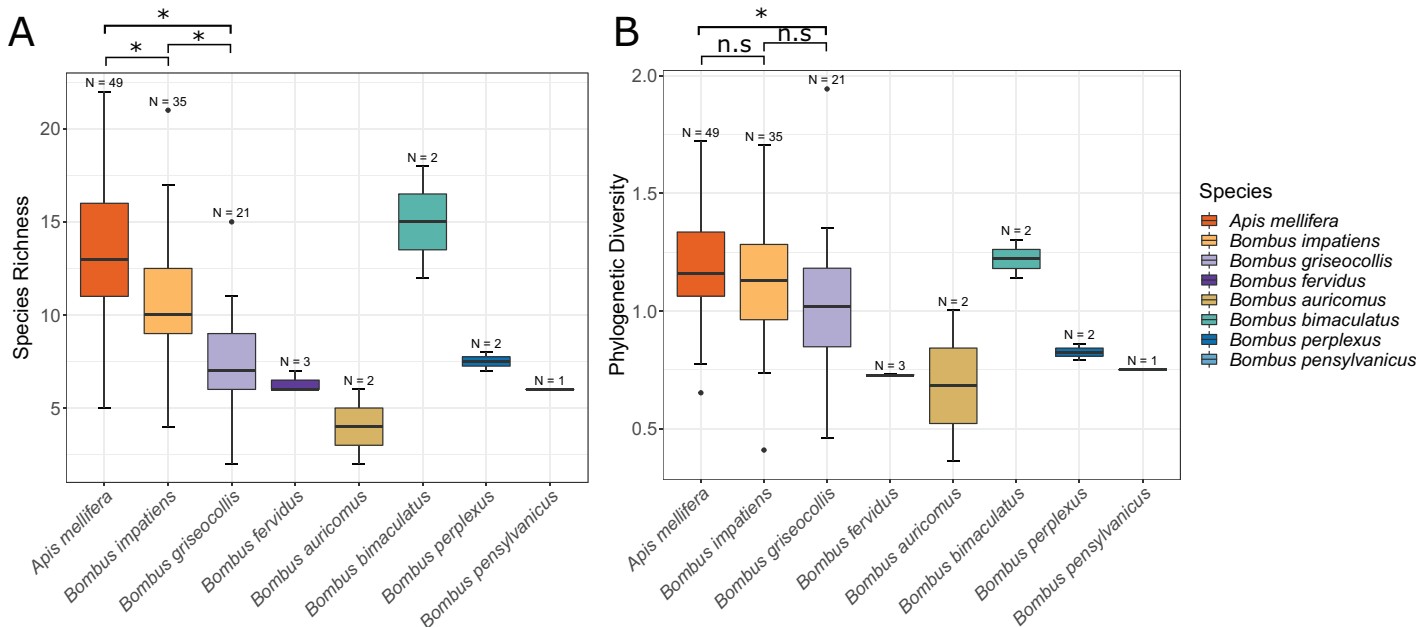

**Figure 3 Boxplot of species richness and phylogenetic diversity in all bee species.** An asterisk (*) indicates Tukey's HSD: $p < 0.05$. (A) All focal species have significantly different bacterial richness from one another. (B) There was no significant difference in phylogenetic diversity between bumble bee species. Honey bees have a significantly higher phylogenetic diversity than *B. griseocollis*. 

ASV 1 (*Gilliamella apicola*) was ubiquitous and was found in 71.3% of bees of all species. It was also one of the most frequently occurring ASVs in each of the focal species (42.9–95.9%) (Fig. 2). *Snodgrassella alvi*, on the hand, were found as distinct sequence variants in different groups of bees. ASV 3 (*Snodgrassella alvi*) was found only in *A. mellifera* and never in bumble bees. For *B. impatiens* and *B. griseocollis*, the two focal bumble bee species, ASV 4 was the dominant *S. alvi*. For most of the other bumble bee species, ASV 5 was the most commonly occurring *S. alvi*, being found in all individuals of *B. auricomus* (two out of two), *B. fervidus* (three out of three) and *B. pensylvanicus* (one out of one). In *Bombus bimaculatus*, ASV 44 was the sole representative of *S. alvi*. None of the *S. alvi* sequence variants found in bumble bees were found in honey bees.

## Bacterial diversity across and within bee species

For both focal and non-focal bee species, the average number of ASVs per species ranged from 4.00–15.00 (8.79 ± 3.84, mean ± SD) (Fig. 3A) with the overall number of ASVs per individual ranging from 2–22 (10.96 ± 4.15). ASV richness differed significantly between all three focal species (Fig. 3A; ANOVA by species: $F_{2,102} = 22.32$, $p < 0.001$; Tukey's HSD: *A. mellifera*–*B. impatiens*: $p = 0.013$, all other Tukey's HSD: $p < 0.001$). Bacterial phylogenetic diversity also differed by species with *A. mellifera* observed to have significantly higher phylogenetic diversity than *B. griseocollis* (Fig. 3B; ANOVA by species: $F_{2,102} = 3.90$, $p = 0.023$; Tukey's HSD: $p = 0.019$). Bacterial phylogenetic diversity of the two focal *Bombus* species were not significantly different (Fig. 3B).
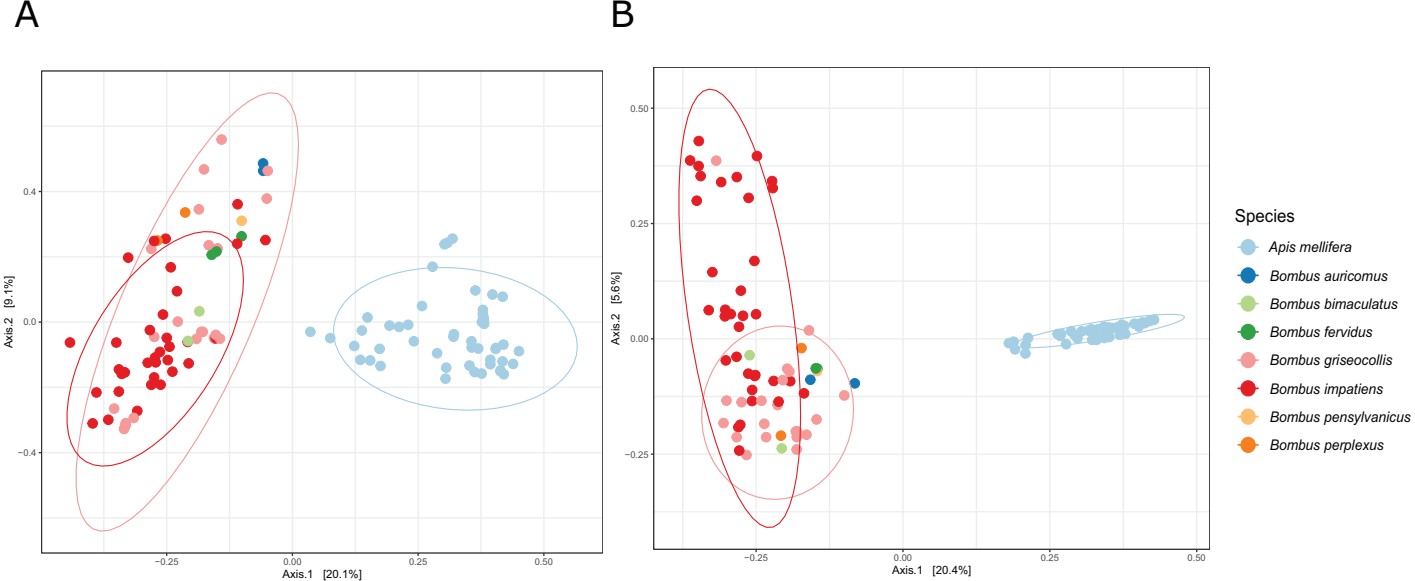

**Figure 4** PCoA plots of individual bee gut microbial communities based on (A) Bray–Curtis distance and (B) Jaccard distance. Individual bees are represented by filled circles and bee species are represented by different colors. Ellipses indicate 95% confidence level grouped by focal species. Bee species have distinct gut microbiome communities. 

Across bee species, we found distinct gut microbiome compositions due to a general lack of sharing of common bacterial ASVs (Figs. 4A and 4B, PERMANOVA Species: Bray–Curtis Pseudo $F_{2,97}$ = 14.98, $R^2$ = 22.02%, $p$ = 0.001; pairwise comparisons: *B. impatiens–B. griseocollis*: $p$ = 0.003, all other pairwise comparisons: $p$ = 0.001; Jaccard Pseudo $F_{2,97}$ = 16.67, $R^2$ = 23.78%, $p$ = 0.001, all pairwise comparisons: $p$ = 0.001). Site differences only explained a moderate amount of bacterial variation (PERMANOVA Site: Bray–Curtis Pseudo $F_{5,97}$ = 1.83, $R^2$ = 6.73%, $p$ = 0.001, pairwise comparisons: Clifton Institute—St. Benedict, Clifton Institute—Residence, & St. Benedict—Residence were not significantly different, all other pairwise comparisons: $p$ < 0.05; Jaccard Pseudo $F_{5,97}$ = 16.67, $R^2$ = 7.08%, $p$ = 0.002, pairwise comparisons: Clifton Institute—St. Benedict & St. Benedict—Residence were not significantly different, all other pairwise comparisons: $p$ < 0.05). PCoA analyses based on Bray–Curtis and Jaccard distances both indicated that the most important differences in microbial communities were between honey bees and bumble bees (Fig. 4, Axis 1).

Bacterial community dispersion, an indication of beta diversity, was lowest in *A. mellifera* compared to *Bombus* species using both Bray–Curtis and Jaccard dissimilarity distances, an expected result given the sharing of common ASVs across individual bees. Bacterial community dispersion was not different between the two *Bombus* species (Figs. 5A and 5B; ANOVA: Bray–Curtis $F_{2,102}$ = 27.19, $p$ < 0.001; Tukey's HSD: *B. griseocollis–B. impatiens* $p$ = 0.834, all other pairwise comparisons: $p$ < 0.001; ANOVA: Jaccard $F_{2,102}$ = 45.36, $p$ < 0.001; Tukey's HSD: *B. griseocollis–B. impatiens* $p$ = 0.204, all other pairwise comparisons: $p$ < 0.001).

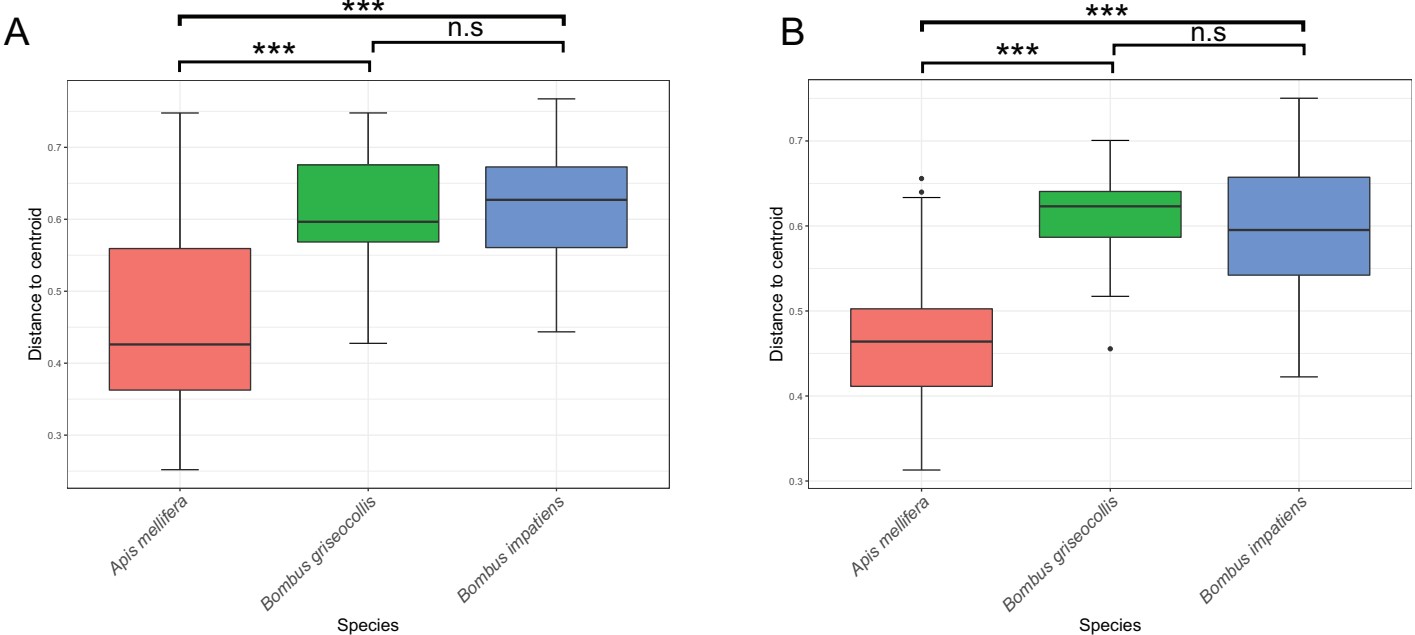

**Figure 5 Microbiome community dispersion between bee species.** Euclidean distances between individual gut microbiota and respective group centroid were calculated using (A) Bray–Curtis and (B) Jaccard distance metrics. Greater dispersion of bacterial gut microbiota in *Bombus* species compared to *Apis*. Three asterisks (***) indicate Tukey's HSD: $p < 0.001$.

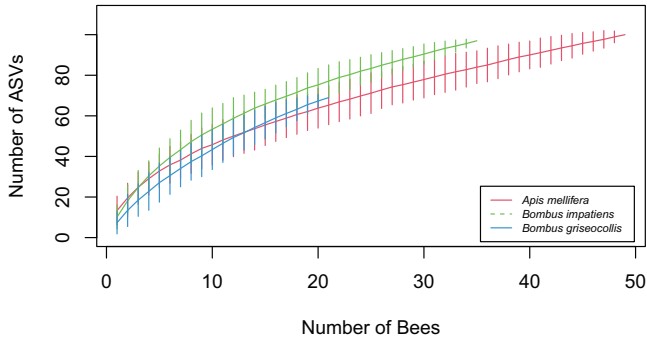

**Figure 6 Bacterial ASV accumulation curves for the three focal bee species.** As more bees were sampled, *B. impatiens* accumulated bacterial ASVs quicker than *B. griseocollis* and *A. mellifera*. A total of 97 ASVs were found in *B. impatiens*, while *A. mellifera* had 100 ASVs, and *B. griseocollis* had 69 ASVs. Error bars represent 95% confidence intervals.

Accumulation curve-based gamma analysis showed that the overall number of gut bacterial ASVs detected was highest in *A. mellifera* ($N = 100$), followed by *B. impatiens* ($N = 97$) and *B. griseocollis* ($N = 69$) (Fig. 6). After limiting the number of bees to 21 (sample size of *B. griseocollis*, the smallest among the focal species), the rarified total ASV count of *A. mellifera* ($65.96 \pm 5.15$) was lower than that of *B. griseocollis* (69) and *B. impatiens* ($77.15 \pm 3.89$). This rarified number of ASVs was highest in *B. impatiens* as a result of its relatively high alpha diversity (number of ASVs detected per bee, Fig. 3A) and a

**Table 3 Bacterial ASV indicators associated with parasitic infection in focal bee species.** Co-infection refers to infection by both Trypanosome and *Vairimorpha*.

| Focal bee species | Infection type | ASV | Phylum | Class | Order | Family | Genus | Species | IndVal score | *p*-value |
|---|---|---|---|---|---|---|---|---|---|---|
| *A. mellifera* | Co-infection | ASV50 | Proteobacteria | Gammaproteobacteria | Enterobacteriales | Enterobacteriaceae | Pluralibacter | NA | 0.739 | 0.002 |
| *B. impatiens* | Uninfected with Trypansomes | ASV13 | Proteobacteria | Gammaproteobacteria | Orbales | Orbaceae | Gilliamella | apicola | 0.647 | 0.017 |
| | Uninfected with Trypansomes | ASV29 | Proteobacteria | Gammaproteobacteria | Pseudomonadales | Moraxellaceae | Acinetobacter | NA | 0.639 | 0.044 |
| | Uninfected with Trypansomes | ASV25 | Bacteroidetes | Flavobacteriia | Flavobacteriales | Flavobacteriaceae | NA | NA | 0.654 | 0.008 |

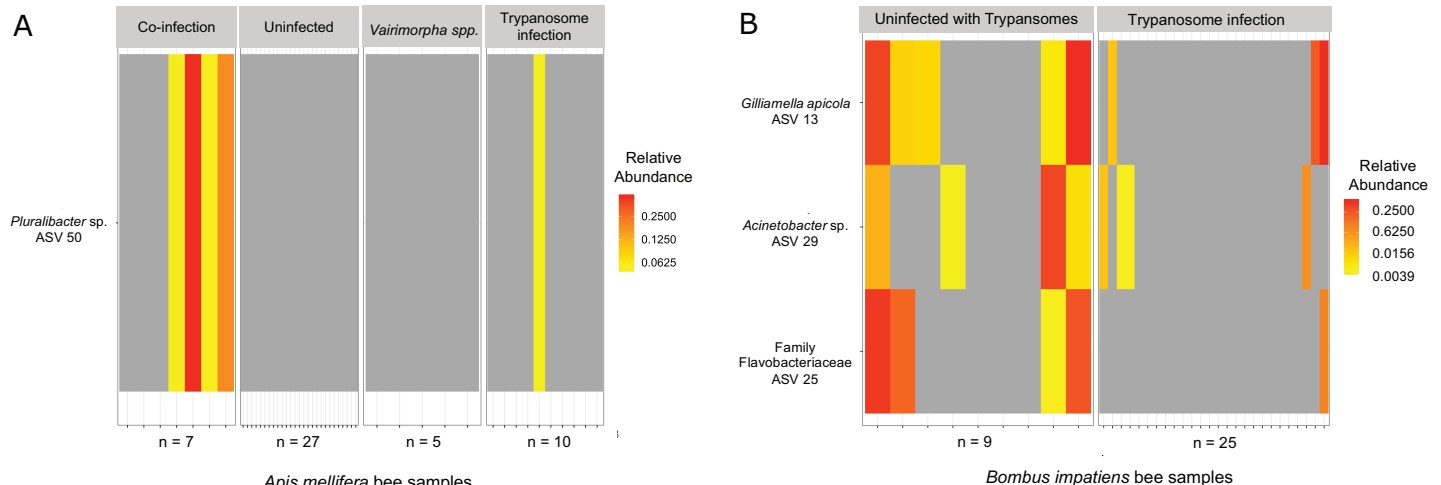

**Figure 7 Specific bacterial taxa associated to infection status.** Using indicator species analysis, (A) we found *Pluralbacter* sp. (ASV 50) abundant in *A. mellifera* with co-infections and, (B) three bacterial taxa (ASV 13, ASV 29, ASV 25) abundant in *B. impatiens* uninfected with Trypanosomes. We found no bacterial taxa associated with infection in *B. griseocollis*.

larger turnover in bacterial ASVs from bee to bee (*i.e.*, high dispersion, Fig. 5A). Despite *B. griseocollis* having the lowest alpha diversity among the three focal species (Fig. 3A), its gamma diversity was similar to *A. mellifera* because the latter has the lowest bacterial community dispersion (Fig. 5A). None of the species accumulation curves plateaued, indicating that our sampling did not fully uncover all gut bacterial ASVs in each of the focal bee species.

## Parasite infection described by indicator bacterial ASVs

Out of 105 focal bees screened, six were found to have *Vairimorpha* infection only, 49 had Trypanosome infection only, 10 were infected by both types of pathogens, and 40 had none (Table 2). We found indicator ASVs associated with infection status in *A. mellifera* and *B. impatiens* (Table 3). In *A. mellifera*, ASV 50 (*Pluralibacter* sp.) associated positively with co-infection by Trypanosomes and *Vairimorpha* spp. (Fig. 7A; IndVal = 0.739; *p* = 0.002). In *B. impatiens*, we found individuals with no infection by Trypanosomes were

characterized by greater abundance of ASV 25 (family assignment Flavobacteriacea), ASV 13 *G. apicola*, and ASV 29 *Acinetobacter* sp. (Fig. 7B).

## DISCUSSION

### Core corbiculate taxa

We investigated bee species (*Apis mellifera and Bombus* spp.) with different life histories and colony sizes and found that they maintained simple gut microbiomes dominated by a small number of core bacterial lineages. The prevalence of three key genera, *Gilliamella*, *Lactobacillus*, and *Snodgrassella*, and specifically the species *G. apicola* and *S. alvi*, across bee species suggests specific and important functions, such as digestion and protection from pathogens are carried out by these bacteria within the bee gut environment (*Kwong & Moran, 2016*). These three bacterial genera are part of the core bacterial phylotypes found across corbiculate bees (*Snodgrassella, Gilliamella, Lactobacillus* Firm-4 and Firm-5, and *Bifidobacterium*), suggesting that their associations started at the beginning of their diversification (*Martinson et al., 2011*; *Koch & Schmid-Hempel, 2011*; *Kwong et al., 2017*).

As we found correspondence in the broad bacterial lineages reported by other studies, the use of amplicon sequence variants in our study afforded better resolution in distinguishing differences in bacterial taxa when compared to methods that clustered reads into OTUs based on sequence similarity thresholds (*Moran et al., 2012*; *Ganeshprasad et al., 2022*). For *G. apicola*, we found ASV 1 was widely shared by both *A. mellifera* (occurred in 96% of individuals) and *Bombus* individuals (occurred in 53% individuals), suggesting that it is a generalist bacterial taxon (*Lim et al., 2015*; *Kwong & Moran, 2016*). However, there was also one *G. apicola* ASV (ASV 7, which differed from ASV 1 by two nucleotide changes) that was found exclusively in *A. mellifera*. In addition to the possibility that ASV 7 is an *A. mellifera* specific ASV, it is also possible that ASV 1 and 7 represent intra-genomic 16S rRNA polymorphism found only in *A. mellifera*. This is because both ASV 1 and 7 mapped with 100% identity to the genome of the type strain of *G. apicola* (GenBank accession number: CP_007445.1, *Kwong et al., 2014*), but to separate rRNA operons. In comparison to *G. apicola*, *Snodgrassella* ASVs showed even stronger host specificity with different sequence variants dominant in different species; *Snodgrassella alvi* ASV4 was dominant in *B. impatiens* and *B. griseocollis*, ASV 3 in *A. mellifera*, and ASV 5 in *B. auricomus, B. pensylvanicus* and *B. fervidus*. In fact, ASV 4 and ASV 5 mapped with 100% identity to 16S rRNA sequences of the type strains of the newly named *S. communis* (GenBank accession: NR_181959.1) and *S. gandavensis* (GenBank accession: NR_181960.1), respectively (*Cornet et al., 2022*). Strains of these two *Snodgrassella* species, along with three other species that are yet to be named, were isolated from various *Bombus* species (*Cornet et al., 2022*). In support of our finding that *Snodgrassella* is more host specific than *Gilliamella*, *Cornet et al. (2022)*, using whole-genome sequences of 75 *Snodgrassella* strains, shows that *Snodgrassella* are not shared between *Bombus* and *Apis*. Another study also found a similar pattern where *Snodgrassella* 16S rRNA haplotypes have stronger host association across various *Bombus* and *Apis* species when compared to *G. apicola* (*Koch et al., 2013*).

We suggest that the difference in the strength of host association of *Gilliamella* and *Snodgrassella* ASVs may relate to the transmissibility of these bacteria and how tightly they interact with the host's gut tissues. *Snodgrassella* is in closer contact with the honey bee ileum, residing adjacent to the gut wall, while *Gilliamella* forms a layer on top of it, in contact with the lumen (*Martinson, Moy & Moran, 2012*; *Kwong et al., 2014*). *Snodgrassella* adhering to the gut wall suggests stronger host-specific interactions, and therefore it is more likely to be transmitted vertically or through oral trophallaxis (*Kwong et al., 2017*; *Cornet et al., 2022*). On the other hand, *Gilliamella* may be transmitted outside the nest environment (*e.g.*, through flowers), although the prevalence of this mode of transmission needs to be verified with more studies (*Koch et al., 2013*; *Graystock, Goulson & Hughes, 2015*; *Sookhan et al., 2021*). The differences in the strength of interactions with host tissues, and the ability to survive outside of the gut environment thus likely explain differences in host specificity, and, potentially, cospeciation patterns between *G. apicola* and *S. alvi*.

*Lactobacillus* was similar in host specificity to *Snodgrassella*. The three most common *Lactobacillus* in honey bees (found in 51.0–100% of bees) were ASV 6 (*L. apis*), ASV 23 (*L. sp.*, aligning with 100% identity to sequences of multiple named species in GenBank (accession numbers: NR_179358.1, NR_12625.1 and NR_126250.1) and ASV 26 (*L. melliventris*), and they were almost never found in bumble bees. *Lactobacillus* species found in bumble bees were similarly not found in honey bees, but the most common one (ASV 12, *L. bombicola*) was found across multiple bumble bee species.

The least abundant of the core bacterial genera was Bifidobacterium. Our results, in agreement with other studies, show that *Bifidobacterium* spp. are a moderate to minor component in both *A. mellifera* and *Bombus* spp. (*Moran et al., 2012*; *Kwong et al., 2017*). The most prevalent *Bifidobacterium* species in each bee genus was found in only 12.2–16.7% of the bees. Together with *Gilliamella* bacteria, *Bifidobacterium* bacteria perform the function of digesting polysaccharides, a function that *Snodgrassella* and *Lactobacillus* do not possess (*Zheng et al., 2019*).

## Non-core and environmental bacterial taxa

We detected some but not all bacterial taxa considered by other studies to be *Apis* or *Bombus* specific (*Kwong et al., 2017*). For example, for *A. mellifera*, we detected ten *Frischella* ASVs but no *Bartonella apis*. It is possible that the time of sample collection had an impact, as *B. apis* is more common during the winter season. This seasonal turnover is likely driven by changes in bee diet, with *B. apis* increasing in abundance when pollen consumption lowers (*Li et al., 2022*). For *Bombus* spp., we found three *Bombiscardovia* ASVs, but none for *Schmidhempelia*. The former bacterial taxa are known *Bombus* symbionts (*Killer et al., 2010*; *Kwong et al., 2017*), and the most common sequence variant (ASV 52) was present in 27.3% of our bumble bee samples.

Some bacterial taxa we detected inside bees were known to be common in outside environments, and thus likely originated from there and have the potential to be transmitted through shared resources such as flowers (*McFrederick et al., 2012*). *Rhizobium* spp. (comprising four ASVs found in 2.0–34.7% of *A. mellifera*) and *Glucnobacter* sp. (ASV 87, found in 5.8% of *Bombus* spp.) are soil dwelling microbes

involved in nitrogen fixation (*Eskin, Vessey & Tian, 2014*; *Ouyabe et al., 2020*). However, each group of bacteria was found exclusively in either honey bees or bumble bees, suggesting little to no transmission of such bacteria between these two groups of bees. Bacteria we found that are commonly associated with flowers and pollen included *Fructobacillus tropaeoli* (ASV 11) (*Endo et al., 2011*), *Saccharibacter floricola* (ASV 9) (*Jojima et al., 2004*), and *Zymobacter palmae* (ASV 114) (*Okamoto et al., 1993*). Interestingly, while these bacteria were common in bumble bees (found in 10.6–31.8% of *Bombus* individuals), they were never found in honey bees. Thus, even for bacteria that originated from outside of bee intestinal tracts, sharing between honey bees and bumble bees can be limited through mechanisms such as floral resource partitioning, host-specific differences (*e.g.*, cuticular hygiene, hairiness) and differences in the condition under which resources are stored (*Keller et al., 2021*).

## Diversity differences between honey bees and bumble bees

When compared to *Bombus* individuals, *A. mellifera* individuals have a higher average alpha diversity. This difference may be explained by the larger size of honey bee colonies, which provides for a larger pool of colonizing bacteria, and more opportunities for transmission through direct contact with nestmates and hive materials (*Martinson, Moy & Moran, 2012*). On the other hand, at the beta diversity level, larger colonies of honey bees may explain a lower turnover of bacterial ASVs among individuals (*i.e.*, lower microbiome dispersion), when compared to *Bombus* individuals. The smaller and spatially dispersed *Bombus* colonies facilitate differentiation of the microbiome among individuals. Additionally, *Bombus* species may have greater variation in their gut microbiome due to their annual life cycle. With only newly mated queens surviving the winter (*Sarro et al., 2021*), as opposed to the perennial colonies of honey bees, the microbiome of entire bumble bee colonies are subjected to stronger stochasticity due to their founding by single individual queens.

Despite honey bees having higher alpha diversity, our study shows that they have lower gamma diversity than the two focal bumble bee species. This shows that the lower turnover of ASVs among honey bee individuals overrides the effects of possessing more ASVs per individual. In addition to having larger colonies, the lower turnover of ASVs among honey bee individuals may be caused by them having a higher flower constancy during foraging bouts (*Grüter & Ratnieks, 2011*; *Leonhardt & Blüthgen, 2012*). By visiting fewer plant species per foraging bout, honey bees may be exposed to fewer environmental bacterial species when compared to bumble bees. Among the focal bumble bee species, *B. impatiens* has higher gamma diversity than *B. griseocollis*. This could be due to *B. impatiens* being able use more habitat types and thus exposed to a greater variety of bacterial species than *B. griseocollis*, which tends to prefer a smaller set of habitats, such as forested landscapes, planted meadows and shrubby habitats (*Novotny et al., 2021*).

## Bacterial ASVs and parasite infection

We found *G. apicola* (ASV 13) in *B. impatiens* positively associated with bees uninfected by Trypanosomes. Previous work on the microbiome of *Bombus* species also reported that

*Gilliamella* spp. are associated with bees uninfected by *Crithidia* (*Cariveau et al., 2014*). *Gilliamella*'s potential protective functions may be explained by its ability to form biofilms that are in contact with the gut lumen (*Martinson, Moy & Moran, 2012*; *Kwong & Moran, 2013*), which provides a barrier against enteric pathogens (*Cariveau et al., 2014*). An alternative explanation to *G. apicola* providing defenses against enteric pathogens is that biofilms formed in the gut by these bacteria may be disrupted by parasitic infections, but the bacteria do not participate in active defenses (*Cariveau et al., 2014*; *Paris et al., 2020*). In either case, it is unclear why other *Gilliamella* ASVs do not associate with a lack of infection in any of the focal bee species, and why the association for ASV 13 only occurred in *B. impatiens* and not in *B. griseocollis* (ASV 13 did not occur in *A. mellifera*). While it is possible that some of these results are driven by differences in sample size, infection rate and statistical power, future studies should investigate whether association with infections is tied to specific *G. apicola* strains or ASVs, or to the bacterial species in general. In addition to *G. apicola*, we found *Acinetobacter* sp. (ASV 29) and a member of Flavobacteriaceae (ASV 25) associated with *B. impatiens* uninfected with Trypanosomes. ASV 25 has >99% sequence similarity with *Apibacter mensalis*, a rare bumblebee gut microbiome first isolated by *Praet et al. (2016)*. Though they are less abundant than taxa such as *Gilliamella* and *Snodgrassella*, both are known gut symbionts of bees, and their association with decreased infection by Trypanosomes has been demonstrated by other studies as well (*Koch & Schmid-Hempel, 2011*; *Mockler et al., 2018*).

In *A. mellifera*, we found gut microbe *Pluralibacter* sp. (ASV 50) associated with co-infection by Trypanosome and *Vairimorpha*. *Pluralibacter* sp. (ASV 50) likely represents an opportunistic microbe taking advantage of a weakened immune system or infections by parasites. ASV 50 has 98.7% similarity to *Pluralibacter gergoviae* (*Harada, Oyaizu & Ishikawa, 1996*), a pathogen found widely in the environment and most closely associated with nosocomial infections (*Brenner et al., 1980*; *Ganeswire, Thong & Puthucheary, 2003*). Consistent with previous studies, we found no gut bacteria that are associated with infection of *Bombus* sp. purely by *Vairimorpha* (*Koch & Schmid-Hempel, 2011*; *Cariveau et al., 2014*).

## CONCLUSIONS

Bee populations are declining largely due to land use changes, widespread use of agrochemicals, infection by pathogens, and climate change (*Potts et al., 2010*; *Goulson et al., 2015*). As studies across the animal kingdom have found that gut microbiome influences host fitness through interactions with the immune system (*Motta, Powell & Moran, 2022*), reproduction (*Antwis et al., 2019*) and nutrition (*Watanabe & Tokuda, 2010*; *Arora et al., 2022*), developing a better understanding of the bacterial gut microbiome of bees and its contributions to the health of individuals and populations will help us devise strategies to mitigate the impacts of some of the abovementioned negative factors (*Bahrndorff et al., 2016*; *West et al., 2019*). One such example is the development of probiotics composed of common gut bacteria, which have been shown to help activate bee immunity genes (*Motta et al., 2022*).

In keeping with findings from other studies, we found that honey bees and bumble bees possess simple gut microbiomes that were dominated by a small number of bacterial taxa. This indicates that gut microbes are closely associated symbionts that perform important functions in their hosts, and whose disruptions can provide important insights into host health. Aside from one generalist bacterial ASV, *G. apicola* (ASV 1), honey bees and bumble bees rarely share gut bacteria, even ones that likely derived from the environment. As expected, given differences in ecology and social structure between honey bees and bumble bees, the former has higher gut bacterial alpha diversity but lower turnover of bacteria among individuals. By combining with results from screening of bees for Trypanosome and *Vairimorpha* parasites, we found that some symbiont bacterial species associated with a lack of infection in bumble bees, while an opportunistic and potentially pathogenic bacterium associated with co-infections in honey bees. We suggest that future studies should investigate whether responses to parasitic infections by gut bacteria are strain specific or shared at the species level. Insights from this type of research can lead to further studies that investigate the efficacy of bee probiotics that are composed of different bacterial strains. Finally, future studies should address how bees that occupy the same habitats maintain different sets of environmental bacteria in their intestinal tracts–whether this is driven by bees visiting different resources or by differences in bee physiology and nest conditions that effect microbial filtering (*Anderson et al., 2014*; *Keller et al., 2021*).

## ACKNOWLEDGEMENTS

David Lambrecht provided data on infection status of bees and T'ai Roulston assisted in making arrangements to visit study sites. Marcus A. H. Chua and Masoumeh Sikaroodi provided help and training for NA in the laboratory. The George Mason University Microbiome Analysis Center provided access to bead-milling equipment needed for DNA extraction. Two anonymous reviewers provided invaluable comments and critiques of the manuscript.

### Funding

George Mason University Department of Biology Research Semester program provided partial funding for the sequencing of 16S rRNA libraries. The funders had no role in study design, data collection and analysis, decision to publish, or preparation of the manuscript.

### Grant Disclosures

The following grant information was disclosed by the authors:
George Mason University Department of Biology.

### Competing Interests

The authors declare that they have no competing interests.

## Author Contributions

- Navolle Amiri performed the experiments, analyzed the data, prepared figures and/or tables, authored or reviewed drafts of the article, and approved the final draft.
- Mia M. Keady analyzed the data, prepared figures and/or tables, authored or reviewed drafts of the article, and approved the final draft.
- Haw Chuan Lim conceived and designed the experiments, performed the experiments, analyzed the data, prepared figures and/or tables, authored or reviewed drafts of the article, and approved the final draft.

## Field Study Permissions

The following information was supplied relating to field study approvals (*i.e.*, approving body and any reference numbers):

Permissions to access study sites were obtained from B. Harris (The Clifton Institute), T. Roulston (Blandy Experimental Farm), C. Vuocolo (Piedmont Environmental Council), L. Edsall (St. Benedict Monastery), G. Perilla (George Mason University) and S. Brandsetter (private residence, Winchester, VA).

## DNA Deposition

The following information was supplied regarding the deposition of DNA sequences:

The datasets generated and analyzed from this study are available at the NCBI SRA repository: PRJNA874076.

## Data Availability

The final files for microbial 16 rRNA analyses (feature table, taxonomy table, phylogenetic tree, and metadata file) and R codes are available at GitHub and Zenodo:

https://github.com/EvoGenLab/honey_bumble_bee_16S

Mia Keady, hawchuan, & EvoGenLab. (2023). EvoGenLab/honey_bumble_bee_16S: Release: 2023-4-13 (Version v1). Zenodo. https://doi.org/10.5281/zenodo.7827260

## Supplemental Information

Supplemental information for this article can be found online at http://dx.doi.org/10.7717/peerj.15501#supplemental-information.

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
