# Peer review of "Honey bees and bumble bees occupying the same landscape have distinct gut microbiomes and amplicon sequence variant-level responses to infections"

_PeerJ, doi:10.7717/peerj.15501_

## Round 0.1 · original submission · Major Revisions

Dear Dr. Amiri and colleagues:

Thanks for submitting your manuscript to PeerJ. I have now received two independent reviews of your work, and as you will see, the reviewers raised some concerns about the research. Despite this, this the reviewers are optimistic about your work and the potential impact it will have on research studying bee microbiomes. Thus, I encourage you to revise your manuscript, accordingly, taking into account all of the concerns raised by both reviewers.

While the concerns of the reviewers are relatively minor, this is a major revision to ensure that the original reviewers have a chance to evaluate your responses to their concerns.

Importantly, please ensure your Materials and Methods are clearly stated. The methods should be clear, concise and repeatable. Other explanations of the results need some clarity. Please elaborate more on your finding and their relevance in the Discussion.

There are many minor suggestions to improve the manuscript (typos, nuances, etc.).

I agree with many of the concerns of the reviewers, and thus feel that their suggestions should be adequately addressed before moving forward.

I look forward to seeing your revision, and thanks again for submitting your work to PeerJ.

Good luck with your revision,

-joe

Reviewer 1 ·

Basic reporting

This manuscript is written professionally and clearly, with only a few small errors present in formatting. The literature is clearly acknowledged throughout the work, and while there are a couple of occasions where a citation would be useful, the depth of research is evident in the reference list. The background to the study is clearly stated, showing how this study fits within a niche in this field.
Raw data have been shared externally, but these are well signposted and are accessible. The manuscript is well structured to the style of a report and follows the author guidelines well. The rationale and the following results are in line with one another.

Experimental design

The study has a clearly defined research question, and data collection methods are well explained in terms of bee collection and microbiome analysis. The knowledge gap that the study fills is clearly explained. I would have liked to see a little more on bee identification (as currently there is a reliance on just one identifier, so reference to identification guides would be useful for future studies). This said, microbiome analysis methods are clearly stated and could be replicated.
There are a couple of points regarding ethics that could be expanded on, such as the ethical approval for the study, and how ethics was taken into account when euthanasing the bees. While bees are not traditionally afforded protection, it would be good to see how ethics was used to inform sample size or euthanasia protocol.

Validity of the findings

The study findings are generally well considered and the results are clear to follow. I would caution the authors to just explain whether normality had been checked for data where parametric tests were used as currently this is not always stated.
The other area to develop is the consideration as to what the results mean. Currently, the interpretation is that the microbiome is reflective of what is symbiotic in the bee. There is limited consideration of what is ubiquitous on shared resources (e.g. flowers) and how this may impact the microbiome. Please consider this in more depth in the discussion. Similarly, a bit more depth on future study directions - and the potential value of this research from an ecological perspective would lend relevance to the study findings.

Additional comments

Overall, this is a carefully considered and professionally formatted study and has merit in terms of microbiome science. I would recommend some small adjustments to the discussion, and clarification regarding the use of ANOVAs.

Annotated reviews are not available for download in order to protect the identity of reviewers who chose to remain anonymous.

Reviewer 2 ·

Basic reporting

Increasing evidence are shown that microbiota plays important roles in gut homeostasis in the way that could affect host health, thus characterizing the composition and difference of gut microbiome for native species would be helpful to better understand their behaviors under perturbations. In this work, Amiri et al. compared the gut microbiome profile across multiple bee species and address their differences. Interestingly, the authors found that the diverse pattern of A. mellifera gut microbiome which could potentially contribute host infection resistance. Overall, the manuscript is well-structured and well-written, thus I would support the publication of this work if the authors could address some issues below.

Major:
Please add some comparative analysis of the result generated in this work with some previously published works. It is fine if the authors find some major difference between the result but would like to see some overall correlation to validate the finding here.

Introduction:
The part about the bee and their ecological role seems a little too tedious. Instead, authors should add more introduction to the role of gut microbiome in bee’s life cycle, including previously reported mechanisms and understudied questions.

Experimental design

Methods:
The benchmarking for quantifying minimal # of reads required for robust ASV identification is important (Figure S1) and I am glad that the authors performed this analysis.

Validity of the findings

Result:
Figure-2: for different samples of the same bee species, please perform hierarchical clustering based on microbiome profile and it would be cool if the authors could identify some potential enterotypes patterns.

Figure-2: Please point out what distance metric are used for the 16S phylogeny on the left side and include the scale bar.

Line-277: considering the low complexity of bee microbiome, it is not surprising that the relative ab. of some ASVs is negative correlated (if ab. of one ASV is getting higher, rest of them should be lower since their sum are 1) and absolute abundance is not practical for bee gut microbiota. Please revise.
Figure-6 and Line-312 to 315: better to use # of ASV pass specific relative ab threshold since the sequencing data here could be subjected to potential sequencing error and other artifacts.

Additional comments

Please correct the terminology usage for strain. For example, in Line 357 to 360, it is not correct to say “it is a generalist bacterial strain” since strain-level information requires whole genome sequencing data to verify. Please use ASV instead.

Please add more description of the script and data in README on Github

Typo in abstract: “larger, perennial hives”

---

## Round 0.2 · accepted · Accept

Dear Dr. Amiri and colleagues:

Thanks for revising your manuscript based on the concerns raised by the reviewers. I now believe that your manuscript is suitable for publication. Congratulations! I look forward to seeing this work in print, and I anticipate it being an important resource for groups studying bee microbiomes Thanks again for choosing PeerJ to publish such important work.

Best,

-joe

Reviewer 1 ·

Basic reporting

Following the oriignal submission of this manuscript, substantial revisions have been made to the discussion - these are in line with the reviewer feedback. This manuscript is written professionally and clearly, with all revision points on proofreading completed. The literature is clearly acknowledged throughout the work and the background to the study is clearly stated, showing how this study fits within a niche in this field.
Raw data have been shared externally, but these are well signposted and are accessible. The manuscript is well structured to the style of a report and follows the author guidelines well. The rationale and the following results are in line with one another.

Experimental design

The study has a clearly defined research question, and data collection methods are well explained in terms of bee collection and microbiome analysis. The knowledge gap that the study fills is clearly explained. Initial concerns on euthanasia and ethics have now been fully addressed.

Validity of the findings

The findings initially used normally distributed tests. Discussions surrounding prior testing to ensure this is the appropriate test have now been inputted. The discussion of the work is now more reflective of what has actually been completed in this study.

Additional comments

This is an interesting an useful study, and the response to initial review comments is demonstrated clearly.

Reviewer 2 ·

Basic reporting

Appreciation to all the authors for making the additional improvements and analysis in revision. I am very happy to see my concerns/suggestions have been fully addressed.

I would recommend this manuscript for publication at PeerJ.

Experimental design

no comment

Validity of the findings

no comment

Additional comments

no comment